# Improving Evolutionary Multi-View Classification via Eliminating Individual Fitness Bias

**Xinyan Liang**[1,2], **Shuai Li**[1], **Qian Guo**[3], **Yuhua Qian**[1]*, **Bingbing Jiang**[4],
**Tingjin Luo**[5], **Liang Du**[1]

[1] Institute of Big Data Science and Industry, Shanxi University
[2] State Key Laboratory of AI Safety, Beijing, 100086
[3] School of Computer Science and Technology, Taiyuan University of Science and Technology
[4] School of Information Science and Technology, Hangzhou Normal University
[5] College of Science, National University of Defense Technology
`{liangxinyan48,LiShuai_liuzhaona,czguoqian}@163.com, jinchengqyh@126.com`
`jiangbb@hznu.edu.cn, tingjinluo@hotmail.com, duliang@sxu.edu.cn`

## Abstract

Evolutionary multi-view classification (EMVC) methods have gained wide recognition due to their adaptive mechanisms. Fitness evaluation (FE), which aims to calculate the classification performance of each individual in the population and provide reliable performance ranking for subsequent operations, is a core step in such methods. Its accuracy directly determines the correctness of the evolutionary direction. That is, when FE fails to correctly reflect the superiority-inferiority relationship among individuals, it will lead to confusion in individual performance ranking, which in turn misleads the evolutionary direction and results in trapping into local optima. This paper is the first to identify the aforementioned issue in the field of EMVC and call it as fitness evaluation bias (FEB). FEB may be caused by a variety of factors, and this paper approaches the issue from the perspective of view information content: existing methods generally adopt joint training strategies, which restrict the exploration of key information in views with low information content. This makes it difficult for multi-view model (MVM) to achieve optimal performance during convergence, which in turn leads to FE failing to accurately reflect individual performance rankings and ultimately triggering FEB. To address this issue, we propose an evolutionary multi-view classification via eliminating individual fitness bias (EFB-EMVC) method, which alleviates the FEB issue by introducing evolutionary navigators for each MVM, thereby providing more accurate individual ranking. Experimental results fully verify the effectiveness of the proposed method in alleviating the FEB problem, and the EMVC method equipped with this strategy exhibits more superior performance compared with the original EMVC method. (The code is available at https://github.com/LiShuailzn/Neurips-2025-EFB-EMVC)

## 1 Introduction

Multi-view classification (MVC) aims to integrate data from multiple views to improve classification accuracy. A large number of methods have been proposed in this field [1–6]. Among them, evolutionary multi-view classification (EMVC) methods differs from traditional approaches where it dynamically fuses different views through the adaptive mechanism of evolutionary algorithms (EA) [7], rather than relying on static modeling. They have obtained important applications like protein

---

*Corresponding Author.

39th Conference on Neural Information Processing Systems (NeurIPS 2025).

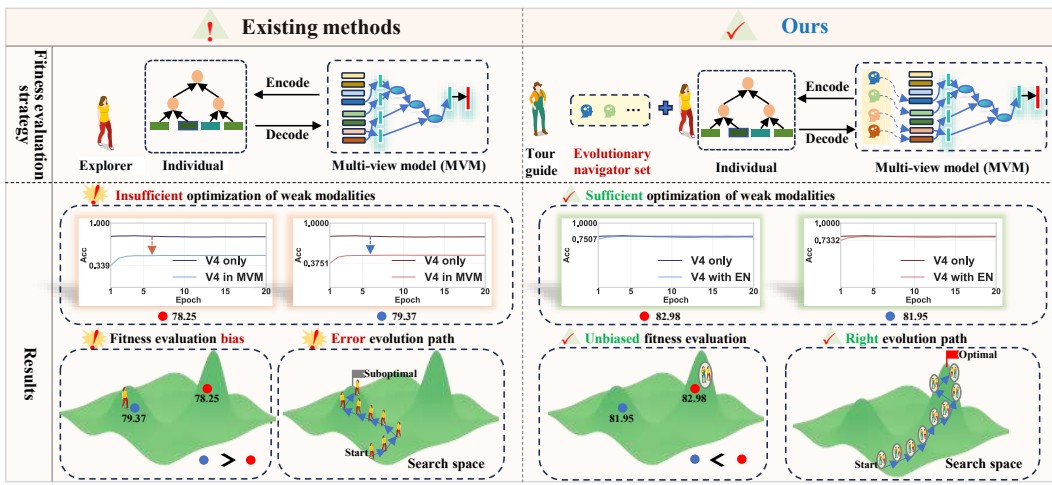

Figure 1: The issues of existing evolutionary multi-view classification and our solutions

secondary structure prediction [8]. Notably, fitness evaluation (FE), as a core step of such methods, runs through the entire process of individual selection, crossover, and mutation. It provides a reliable basis for performance ranking for subsequent operations, and thus its evaluation accuracy directly determines the correctness of the evolutionary direction.

However, when FE fails to correctly reflect the superiority-inferiority relationship among individuals, it will cause confusion in the ranking of individual performance, which in turn misleads the evolutionary direction and traps the process in local optima. This paper is the first to identify the above-mentioned issue in the field of EMVC and formally define it as fitness evaluation bias (FEB). FEB does not generally refer to any error in FE; instead, it specifically refers to the kind of bias that is sufficient to disrupt the true performance ranking of individuals. Given the core role of FE in the entire EMVC method, the occurrence of the FEB issue will in turn lead to systematic distortion in the whole evolutionary process. Specifically, in the selection phase, such biases will interfere with the screening of high-quality individuals, resulting in the abnormal loss of superior genes; in the crossover operation, due to the lack of accurate individual ranking relationships as guidance, it is difficult to form efficient gene combinations; mutation strategies, for lack of reliable guidance, degenerate into random perturbations. More seriously, the iterative optimization characteristic of EA will lead to the continuous accumulation of this bias, which in turn misleads the evolutionary direction and eventually causes the algorithm to fall into local optima (as shown in Fig. 1).

FEB may be caused by a variety of factors, and this paper approaches the issue from the perspective of view information content: in the multi-view model (MVM) decoded from individuals, there are significant differences in the information content among different views. Although the joint training strategy commonly adopted in existing methods can make full use of key information in views with high information content, it fails to fully explore key information in views with low information content. This leads to suboptimal utilization of multi-view data, which in turn causes MVM to fail to achieve optimal performance during the convergence process [9, 10]. Consequently, FE cannot accurately reflect the performance ranking of individuals, thereby triggering FEB. This phenomenon can be analogous to a mountaineer misjudging the highest point due to cognitive bias toward the terrain: if the mountaineer chooses a path based only on a local perspective (partial view information), they may stop prematurely at a suboptimal mountain top, thus missing the main peak that represents the global optimum. To our knowledge, none of the existing EMVC methods have considered this issue, which limits their performance.

To address the issue, we propose an evolutionary multi-view classification via eliminating individual fitness bias (EFB-EMVC) method, which alleviates the FEB problem by equipping each MVM with evolutionary navigators (ENs). EN realizes the directional optimization of view branches based on the knowledge distillation mechanism. Meanwhile, recent studies [11] have pointed out that cross-category information also plays a key role in the distillation process. Inspired by this, we further design EN to distill both corresponding category and cross-category information simultaneously,

so as to better guide the optimization of view branches. Specifically, we propose a wasserstein distance-oriented loss function (WDOLF) and construct a dual-path optimization mechanism to alleviate FEB: (1) MVM optimization ensures the consistency of the overall training objective; (2) the view distillation process fully explores the rich corresponding category and cross-category information in EN, thereby utilizing multi-view data more comprehensively. This mechanism can provide more accurate individual performance ranking, and based on this, calibrate the evolutionary direction in real time to guide the EA to iterate along a more correct path. This process is similar to an explorer moving forward under the guidance of a guide who is familiar with the terrain: with each step taken, the explorer can promptly obtain the correct route leading to the highest peak, thus avoiding falling into local optima (as shown in Fig. 1 for details). By virtue of the concise and efficient strategy of "global control + local in-depth exploration", the potential of EMVC methods is fully unleashed. Notably, EN can be seamlessly integrated with other EMVC methods, demonstrating strong generality. The core contributions of this paper are summarized as follows:

1. In the field of evolutionary multi-view classification (EMVC), we are the first to identify and formally define the problem of fitness evaluation bias (FEB), and systematically analyze its impact on the entire evolutionary process.

2. To alleviate FEB, we equip multi-view model with evolutionary navigators (ENs) to fully explore the rich corresponding category and cross-category information therein, thereby achieving efficient utilization of multi-view data.

3. Experimental results fully verify the effectiveness of the proposed method in alleviating FEB. Moreover, EN can be seamlessly integrated into other EMVC methods, demonstrating strong compatibility.

## 2 Related Work

**Evolutionary Multi-View Classification (EMVC):** The EMVC method utilizes the adaptive mechanism of evolutionary algorithms (EA), enables continuous iterative optimization, and ultimately screens out satisfactory multi-view model (MVM) from the population. EDF [7] initializes the population in the search space composed of views and basic fusion operators, and gradually searches for satisfactory MVM through operations including fitness evaluation, selection, crossover, and mutation, pioneering research in this field. However, due to its inherent drawbacks such as long computation time, a series of improvement works have been initiated. For example, DC-NAS [12] accelerates the search process by adopting a divide-and-conquer strategy for data; CSG-NAS [13] and CoMO-NAS [14] further reduce complexity through the core structure search space, significantly improving algorithm efficiency; KS-NAS [15] reduces computational costs via introducing a dynamic knowledge base. As research advances, the credibility issue of MVM has gradually attracted attention. For instance, TEF [16] uses EA as a generator for high-quality pseudo-views, thereby further enhancing the credibility of MVM. Despite the fruitful achievements, the existing EMVC methods have not taken the problem of fitness evaluation bias (FEB) into account, which limits their performance. In contrast, the EFB-EMVC method proposed in this paper effectively alleviates the FEB problem by evolutionary navigators (ENs) into each MVM.

**Knowledge Distillation (KD):** The core idea of KD is to alleviate the challenges of complex models in deployment by transferring knowledge from the teacher model to the lightweight student model [17]. In this study, we draw on the basic idea of KD and introduce it into the EMVC framework, but its implementation relies on the EN. Through the EN, we can transfer knowledge in a targeted manner to the corresponding view branches of the MVM to assist in its training. Unlike traditional KD, the focus of this study is not on proposing a new distillation paradigm, but on using EN to alleviate the inherent FEB in the EMVC framework. Recent work [11] found that the cross-category information contained in the teacher model plays a key role in knowledge transfer. Inspired by this, the EN designed in this study not only provides corresponding category information, but also transmits cross-category information to better guide the optimization of each view branch. It is noted that the EN plays a core role in the process of alleviating FEB, while KD only serves as the implementation mechanism for knowledge transfer.

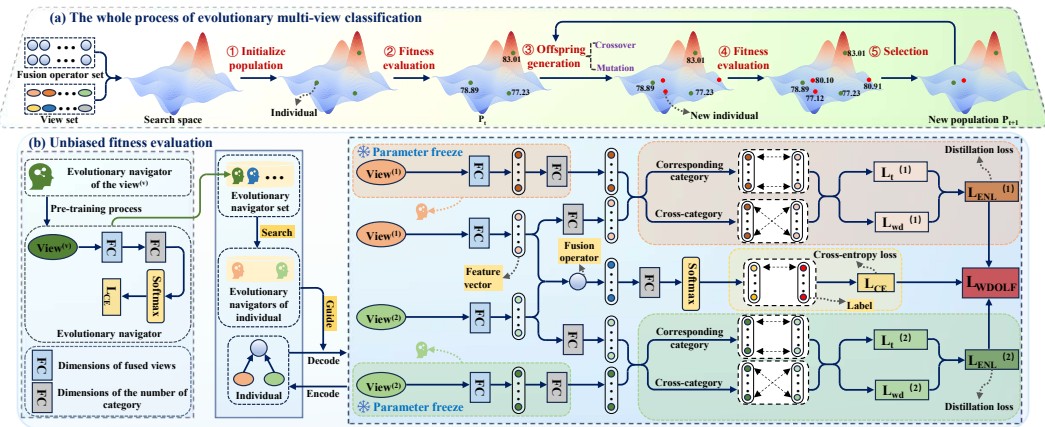

Figure 2: The overview of EFB-EMVC

## 3 Proposed Method

The proposed EFB-EMVC follows the process of the existing evolutionary multi-view classification (EMVC) method. The only difference is that EFB-EMVC alleviates fitness evaluation bias (FEB) by introducing evolutionary navigators (ENs) for each multi-view model (MVM) while existing methods do not. Our EFB-EMVC overview is illustrated in Fig. 2.

### 3.1 The Alleviation of Fitness Evaluation Bias

In the field of EMVC, we are the first to identify and formally define the FEB issue, which exerts a systematic impact on the entire evolutionary process. To alleviate FEB, we equip each view branch in MVM with an EN and guide it by drawing on the knowledge distillation (KD) mechanism. However, traditional KD can only model corresponding category, while recent studies [11] have shown that rich cross-category information is crucial for the distillation process. Inspired by this, we design EN to distill both corresponding category and cross-category information simultaneously, thereby guiding the optimization of view branches more effectively.

Specifically, the EN is defined as a pre-trained teacher model. Taking the $v$-th view as an example: First, the EN corresponding to this view is decoded into a two-layer fully connected deep neural network (DNN), where the dimension of the first layer is the fused view dimension, the dimension of the second layer is the number of categories $C$, and a softmax layer is attached at the end of the network. Subsequently, the training data and test data are input into the DNN to complete the training and testing processes. After training, the optimal DNN parameters are frozen. For a given dataset, we pre-train ENs for all views and store the results in an EN set for subsequent retrieval. During the individual training process, according to the views corresponding to leaf nodes, the ENs required by the individual are retrieved from the EN set to serve as the knowledge source for that individual. The above process is shown in Fig. 2 (b) and is achieved via three steps.

**Step 1** aims to extract the category interrelations from the EN. Taking the $v$-th view as an example: ❶ Obtain the predicted probability distribution outputs of all the training data by passing them into the corresponding EN; ❷ Divide the training data into $C$ category groups according to the resulting predicted probability distribution outputs; ❸ For the $c$-th category divided by the EN, assuming it contains $b_c$ training samples, we collect their logits, and then concatenate them into a feature matrix $X_v^c \in \mathbb{R}^{C \times b_c}$; ❹ Construct the kernel matrix $K_v^c = X_v^c(X_v^c)^\top \in \mathbb{R}^{C \times C}$. Notably, in some datasets, the EN corresponding to certain views may end up classifying fewer than $C$ categories — this indicates that the EN judges there to be no samples belonging to some categories. For the sake of convenient description, we uniformly consider the number of its classified categories as still $C$. For the aforementioned categories with no corresponding samples, the elements in their kernel matrices are replaced with extremely small positive numbers; ❺ After obtaining the kernel matrices $\{K_v^c\}_{c=1}^C$ for all categories, we measure the category interrelations using the HSIC score, and then construct the mutual information matrix $M_v$ for the $v$-th view as follows: for any two elements $K_v^{c_1}$ and

$K_v^{c_2}$ from $\{K_v^c\}_{c=1}^C$, one computes $\text{HSIC}(K_v^{c_1}, K_v^{c_2}) = \frac{1}{C^2}\,\text{tr}\left(\tilde{K}_v^{c_1}\tilde{K}_v^{c_2}\right)$, where $\tilde{K}_v^{c_1} = HK_v^{c_1}H$, $\tilde{K}_v^{c_2} = HK_v^{c_2}H$, and $H = I - \frac{1}{C}\mathbf{1}\mathbf{1}^\top$, with $I$ as the identity matrix and $\mathbf{1}$ a column vector of ones. The resulting mutual information matrix quantifies the statistical correlations between categories in the $v$-th view, where larger values indicate stronger associations between their feature distributions.

**Step 2** aims to effectively guide the training of each view branch by leveraging EN, thereby alleviating FEB. To this end, we have designed an optimization objective referred to as EN guidance loss (ENL). The core idea of ENL is as follows: during the training process, each view branch not only needs to learn information of their corresponding category from EN, but also should capture knowledge of cross-category relationships provided by EN—this enables the achievement of better optimization effects. The ENL loss function consists of two components: The first component is the cross-entropy loss between a view branch and EN with respect to the target category; The second component is the wasserstein distance (WD) between the output distributions of the two (i.e., a view branch and EN) with respect to non-target categories.

The loss function for the first part is $L_t = -(p^{\mathcal{T}})_t \cdot \log(p^{\mathcal{S}})_t$, where $t$ denotes the target category, $p^{\mathcal{T}} = \text{softmax}(z^{\mathcal{T}})$ and $p^{\mathcal{S}} = \text{softmax}(z^{\mathcal{S}})$ represent the output probability distributions of the EN and the view branch. $z^{\mathcal{T}}$ and $z^{\mathcal{S}}$ denote the logits outputs of the EN and the view branch, respectively.

The second part is to calculate the WD between EN and view branches with respect to the probability distributions of non-target categories. For two discrete distributions $P$ and $Q$, WD is defined as:

$$W(P,Q) = \min_{\gamma \in \Gamma} \sum_i \sum_j \gamma_{ij} \cdot c_{ij}, \tag{1}$$

where $\sum_j \gamma_{ij} = p_i$, $\sum_i \gamma_{ij} = q_j$, and $\gamma_{ij} \geq 0$. Here, $\gamma_{ij}$ denotes the transport plan from $p_i$ to $q_j$, $c_{ij}$ is the transportation cost, and $\Gamma$ is the set of feasible couplings that satisfy the marginal constraints.

Since both the EN and the view branch output discrete probability distributions over non-target categories, we adopt the discrete WD formulation. Specifically, we first mask the target category in the logits of both models to obtain the non-target feature vectors $\hat{z}^{\mathcal{T}}$ and $\hat{z}^{\mathcal{S}}$, and apply the same masking to the mutual information matrix to derive the masked mutual information matrix $\hat{M}_v$. Then, we apply temperature-scaled softmax to $\hat{z}^{\mathcal{T}}$ and $\hat{z}^{\mathcal{S}}$, yielding the non-target categories probability distributions $\hat{p}^{\mathcal{T}}$ and $\hat{p}^{\mathcal{S}}$.

To efficiently approximate WD, we employ the entropy-regularized Sinkhorn algorithm[18]. The kernel matrix is defined as $K = \exp(-\hat{M}_v/\epsilon)$, where $\epsilon = 0.05$ is the regularization parameter. The iteration matrix $U$ is initialized as: $U = \frac{1}{C-1} \cdot \mathbf{1}$. Then, the updates are performed iteratively as:

$$V \leftarrow \frac{\hat{p}^T}{K^\top U}, \quad U \leftarrow \frac{\hat{p}^S}{KV}, \tag{2}$$

and after 10 iterations, the optimal transport plan is obtained as:

$$\gamma^\star = \text{diag}(U)\,K\,\text{diag}(V). \tag{3}$$

The WD is then computed as:

$$L_{wd} = \sum_i \sum_j \gamma_{ij}^\star \cdot (\hat{M}_v)_{ij}. \tag{4}$$

Finally, the ENL loss is defined as:

$$L_{ENL} = \alpha L_t + \beta L_{wd}, \tag{5}$$

where $\alpha$ and $\beta$ are hyperparameters used to balance the contributions of the target-category cross-entropy loss and the non-target categories WD.

**Step 3** aims to combine the proposed ENL loss function with the traditional cross-entropy loss to jointly optimize the MVM and its individual view branches. This combined loss not only ensures the consistency of the overall training objective but also makes full use of the critical information from each view, thereby effectively alleviating the FEB problem.

When considering $\mathcal{V}$ views, the whole loss function WDOLF is formulated as follows:

$$L_{WDOLF} = \gamma_1 L_{CE} + \gamma_2 \sum_{v=1}^{\mathcal{V}} L_{ENL}^{(v)} = \gamma_1 L_{CE} + \gamma_2 \alpha \sum_{v=1}^{\mathcal{V}} L_t^{(v)} + \gamma_2 \beta \sum_{v=1}^{\mathcal{V}} L_{wd}^{(v)}, \tag{6}$$

where $L_{CE}$ represents the cross-entropy loss between the output of the MVM and the ground-truth labels, and $L_{ENL}^{(v)}$ represents the ENL loss between the $v$-th view branch of the MVM and the corresponding EN.

### 3.2 The EMVC Method Driven by Unbiased Fitness Evaluation

The core steps of the EFB-EMVC method include population initialization, unbiased fitness evaluation, offspring generation, and selection.

**Population initialization:** First, in the search space composed of views and basic fusion operators, randomly initialize a population $P$ consisting of $k$ individuals encoded in the form of binary trees, where the leaf nodes consist of views and the branch nodes consist of basic fusion operators. In this paper, the basic fusion operators consists of concatenation, addition, multiplication, maximize and average [7]. For each individual, if the binary tree contains $\mathcal{V}$ views, then it must contain $\mathcal{V} - 1$ basic fusion operators. Each individual corresponds to MVM.

**Unbiased fitness evaluation:** First, according to the views corresponding to the leaf nodes of an binary tree, the ENs required by the individual are retrieved from the EN set of the current dataset. Next, the individual is decoded into an MVM. In this MVM, each view first passes through a fully connected layer (whose dimension is set to the dimension of the fused view). Subsequently, the data is processed through two streams: 1) one stream enters a fully connected layer (with the dimension equal to the number of categories) to generate a logits vector, and then the $l_{ENL}$ distillation loss between this vector and the logits output by the EN corresponding to the view is calculated; 2) the other stream is fused with other views via a fusion operator, then passes through a fully connected layer (with the dimension equal to the number of categories) to generate the logits vector of the MVM, and finally enters the softmax layer to obtain a probability distribution, with which the cross-entropy loss is calculated against the true labels. Finally, all losses are combined into the $l_{WDOLF}$ loss function, which is used to optimize the entire MVM and each view branch. Finally, the test accuracy at the optimal state during training is taken as the unbiased fitness value of the individual. The above process is detailed in Fig. 2 (b).

**Crossover and mutation:** The crossover rate is $r = r1$, and the mutation rate is $r = r2$. At this stage, following the principles of traditional evolutionary algorithms, the probability of the crossover operation is significantly higher than that of the mutation operation. For the crossover operation, a non-root node is first randomly selected from each of two randomly chosen binary trees to serve as the crossover point. The selected node along with its subtree is then pruned from the original tree and exchanged with the counterpart from the other tree. These subtrees are subsequently grafted onto the parent node of the original crossover point in the opposite tree, thereby generating two new individual structures. For the mutation operation, a node is randomly selected from the individual's tree structure. If the selected node is a fusion operation node, a new fusion operation is randomly sampled from a predefined list to replace the original operation. If the selected node is a view node, a new view is randomly chosen from a set of views to replace the original one.

---

**Algorithm 1** EFB-EMVC Framework

---

1: **Input:** $D = (X, Y)$: training dataset; $\hat{D} = (\hat{X}, \hat{Y})$: test dataset; $F$: a set of basic fusion operators; $T$: iteration number; evolutionary navigator set;
2: **Output:** The satisfactory MVM and its corresponding accuracy;
3: **Population initialization:** Generate an initial population $P_0$;
4: **Fitness evaluation:** Obtain the unbiased fitness values of all individuals in $P_0$;
5: **for** $t = 1$ **to** $T$ **do**
6:     Generate offspring $Q_t$ using the crossover operator;
7:     Conduct mutation on each individuals in $Q_t$;
8:     Obtain the unbiased fitness values of all individuals in $Q_t$;
9:     Select next generation population $P_{t+1}$ from $Q_t \cup P_t$ using a selection operator;
10: **end for**
11: **return** The satisfactory MVM and its corresponding accuracy.

---

**Selection:** Adopt the binary tournament selection method. Specifically, the selection operation has three application scenarios: (1) Crossover phase: Randomly select two individuals, compare their

fitness values, and retain the better one as Parent 1. Repeat this operation to select Parent 2. Then, perform the crossover operation on the two parent individuals; (2) Mutation phase: Randomly select two individuals and directly choose the one with the better fitness value for mutation; (3) Generating the next generation population: Merge the current population with the offspring population, randomly select two individuals, and retain the better one to be added to the next generation. Repeat this operation $K$ times.

The algorithm framework of the EFB-EMVC method is shown in Algorithm 1.

# 4 Experiments

In this section, we aim to verify the effectiveness of EFB-EMVC method from four aspects: (1) Comparison with SOTA methods; (2) Impact analysis of the evolutionary navigator (EN) on EFB-EMVC; (3) Impact analysis of the distillation loss function ENL on EFB-EMVC; (4) Generality of evolutionary navigator. In addition, we conducted an in-depth analysis of the hyperparameters of EFB-EMVC and the experimental results are presented in Appendix A.4.

## 4.1 Experimental Setup

**Datasets.** In the experiments, nine multi-view datasets are used and they are MVoxCeleb [16], YoutubeFace [19], NUS-WIDE-128 (NUS) [20], Reuters [21], CB [7], MM-IMDB [22], NTU RGB-D [23], and EgoGesture [24]. For the Reuters dataset, two variants named Reuters5 and Reuters3 are generated by adding Gaussian noise [16]. Their detailed descriptions can be found in Appendix A.3.

**Evaluation metrics.** In the experimental process, to effectively avoid the randomness interference caused by data partitioning and network initialization operations, a 5-fold cross-validation strategy was adopted for the MVoxCeleb [16], YoutubeFace [19], NUS-WIDE-128 (NUS) [20], Reuters5 [16] and Reuters3 [16] datasets, dividing each dataset into a training set and a test set. The specific partitioning ratio is as follows: 80% of the samples are used for model training, and 20% are used for model testing. For the remaining datasets, since their original authors have already completed the partitioning of the training and test sets, the experiments were independently repeated five times. The final experimental results are presented in the form of average performance metrics and standard deviations to intuitively reflect the statistical stability of the model performance. For the MM-IMDB dataset, considering the existence of class imbalance, the weighted F1-score is selected as the evaluation metric; a higher value of this metric indicates better model performance. For the remaining datasets, accuracy is used as the evaluation metric.

## 4.2 Experimental Results

**Comparisons with SOTA methods.** In this section, we aim to validate the EFB-EMVC's effectiveness by comparing with four kinds of SOTA methods. They are (1) fixed fusion operators-based methods including addition, average, max, multiplication, concatenation, MLB [25], MFB [26], TFN [27], LMF [28], and PTP [29]; (2) trustworthy multi-view classification methods including TMC [30], TMOA [31], ETMC [32], RCML [33]; (3) traditional adaptive multi-view classification methods including BV [7], SSV [7], MR [7], EmbraceNet [34], AWDR [35], RMAR [36]; (4) EMVC methods including EDF [7], DC-NAS [12], CoMO-NAS [14], CSG-NAS [13] and TEF [16].

The results are shown in Table 1. Based on these results, one can observe that: (1) Compared with the fixed fusion strategy, the EMVC methods demonstrate significant performance advantages on all datasets. This is attributed to its acquisition of a high-performance multi-view model (MVM) through adaptive selection. However, traditional multi-view classification methods, due to being restricted by manually designed features, sometimes have lower performance than that of the fixed fusion strategy. At the same time, the trusted multi-view classification method, because it adopts the way of late-stage desision fusion, results in insufficient feature interaction, and its performance sometimes cannot surpass that of the fixed fusion strategy. (2) Compared with existing EMVC methods, it can be observed that: although these methods outperform other types of methods in terms of performance metrics, the existence of fitness evaluation bias (FEB) causes such methods to fall into local optima, thereby preventing their performance potential from being fully unleashed. In contrast, EFB-EMVC effectively alleviates the FEB issue by introducing ENs into MVM, ensuring that the evolutionary algorithm (EA) can iterate and optimize along a relatively more correct direction to find a more

Table 1: Accuracy comparison results with SOTA methods (mean ± standard deviation), where the best performance is highlighted in bold.

| Methods | MVoxCeleb | YoutubeFace | NUS | Reuters5 | Reuters3 | CB |
|---|---|---|---|---|---|---|
| Add | 87.53±0.41 | 82.40±0.23 | 72.81±0.70 | 79.70±0.25 | 83.46±0.28 | 87.16±0.17 |
| Mul | 72.31±0.90 | 83.18±0.14 | 64.58±0.63 | 77.02±0.38 | 81.89±0.70 | 80.87±1.16 |
| Cat | 87.98±0.20 | 83.05±0.56 | 72.32±0.50 | 79.91±0.28 | 83.66±0.17 | 86.58±0.11 |
| Max | 81.57±0.41 | 81.49±0.29 | 71.36±0.47 | 80.02±0.20 | 84.01±0.28 | 84.21±0.11 |
| Avg | 87.27±0.33 | 82.23±0.17 | 73.00±0.51 | 79.69±0.30 | 83.58±0.28 | 87.05±0.20 |
| MLB (ICLR17) | 87.11±0.67 | 85.20±0.28 | 70.60±0.29 | 80.16±0.15 | 83.80±0.28 | 82.38±0.32 |
| MFB (TNNLS18) | 85.23±0.20 | 82.85±0.17 | 71.34±0.40 | 79.28±0.21 | 83.25±0.18 | 87.94±0.32 |
| TFN (EMNLP17) | 57.53±0.92 | 81.33±0.19 | 63.66±1.22 | 79.95±0.30 | 83.73±0.31 | 73.45±0.30 |
| LMF (ACL18) | 89.92±0.25 | 85.58±0.22 | 71.74±0.70 | 80.03±0.15 | 83.75±0.28 | 82.81±0.18 |
| PTP (NeurIPS19) | 88.61±0.36 | 85.18±0.30 | 71.83±0.50 | 80.10±0.10 | 84.06±0.20 | 85.08±0.11 |
| TMC (ICLR22) | 73.13±0.15 | 71.18±2.27 | 72.73±0.30 | 79.60±0.56 | 84.23±0.35 | 77.87±0.22 |
| TMOA (AAAI22) | 84.72±0.21 | 84.35±0.25 | 72.60±0.48 | 79.11±0.43 | 84.19±0.27 | 86.80±0.10 |
| ETMC (TPAMI23) | 88.70±0.15 | 79.63±1.89 | 73.05±0.67 | 79.80±0.41 | 84.25±0.42 | - - |
| RCML (AAAI24) | 80.51±0.41 | 81.95±0.20 | 72.53±0.55 | 81.39±0.18 | 85.88±0.29 | - - |
| BV (TEVC2021) | 63.25±0.14 | 82.01±0.18 | 68.69±0.59 | 80.61±0.25 | 83.98±0.14 | 77.08±0.15 |
| SSV (TEVC2021) | 85.10±0.23 | 84.43±0.31 | 63.70±0.64 | 79.51±0.41 | 84.71±0.22 | 87.02±0.13 |
| MR (TEVC2021) | 79.92±0.29 | 84.78±0.21 | 64.39±0.85 | 78.24±0.45 | 84.17±0.19 | 83.36±0.21 |
| EmbraceNet (IF19) | 81.74±0.34 | 80.90±1.04 | 72.43±0.38 | 80.07±0.21 | 83.58±0.25 | 85.85±0.09 |
| AWDR(PR19) | 91.08±0.09 | 85.11±0.15 | 72.44±0.66 | 79.69±0.27 | 83.32±0.32 | 86.66±0.16 |
| RMAR(INS22) | 91.54±0.11 | 85.21±0.17 | 72.51±0.67 | 79.84±0.25 | 83.48±0.25 | 85.36±0.46 |
| EDF (TEVC2021) | 93.09±0.20 | 85.83±0.08 | 74.73±0.45 | 81.12±0.25 | 85.49±0.21 | 88.55±0.20 |
| CoMO-NAS (ACMMM24) | - - | - - | 74.24±0.29 | - - | - - | 88.69±0.38 |
| CSG-NAS (IJCAI24) | - - | - - | 74.52±0.40 | - - | - - | 89.20±0.06 |
| DC-NAS (AAAI24) | 92.19±0.07 | 85.28±0.14 | 74.35±0.58 | 81.35±0.28 | 85.86±0.14 | 88.52±0.13 |
| TEF (ICLR25) | 92.41±0.12 | 86.02±0.41 | 75.12±0.57 | 82.26±0.23 | 86.49±0.10 | - - |
| **EFB-EMVC (ours)** | **94.82±0.12** | **87.67±0.17** | **75.79±0.76** | **82.66±0.24** | **86.51±0.12** | **89.67±0.18** |

satisfactory solution. For instance, on the MVoxCeleb, YouTubeFace, and NUS datasets, the accuracy of EFB-EMVC is improved by approximately 1.73%, 1.65%, and 0.67% respectively compared with the suboptimal model. These results fully verify the effectiveness of this mechanism.

To further explore its performance, we carried out additional experiments on other datasets that are often used to validate the effectiveness of the EMVC methods. In this experiment, the SOTA methods compared with EFB-EMVC are roughly divided into two categories: (1) Single-view classification methods, such as Maxout MLP [37], VGG Transfer [38], Inflated ResNet-50 [39], Co-occurrence [40], ResNext-101 [41]; (2) Multi-view classification methods, such as Two-stream, GMU [22], CentralNet [42], MFAS [43], VGG-16 + LSTM [44], C3D + LSTM + RSTTM [45], I3D [46], MMTM [47], MTUT [48], 3D-CDC-NAS2 [49], BM-NAS [50], DC-NAS [12], CoMO-NAS [14], CSG-NAS [13], and MCTS-CSG [51] and HF-MNAS [52]. According to the results in Table 2, compared with single-view classification methods, multi-view classification methods generally achieve better performance. This is mainly due to their ability to fully utilize the rich information across different views, thereby achieving performance breakthroughs. Compared with SOTA methods, EFB-EMVC achieves the best performance on most datasets and also reaches performance comparable to SOTA methods on the EgoGesture dataset. This advantage stems from the EN mechanism of EFB-EMVC — this mechanism corrects the evolutionary direction in real time by alleviating the FEB issue.

**Impact analysis of the evolutionary navigator (EN) on EFB-EMVC.** In this experiment, we aim to analyze the impact of EN on EFB-EMVC (i.e., whether EN alleviates the problem of FEB?) by an ablation analysis on YouTubeFace dataset. To ensure the generality of this method, we took a population consisting of 15 randomly initialized individuals as the research object.

As shown in the left side of Fig. 3, the accuracy of all individuals after applying the EN exhibits a consistent improvement. It is noticed that its core value lies in the reconstruction of the performance order relationship on which the EMVC method relies. The selection mechanism of the EMVC method essentially depends on the performance order relationship to identify high-quality solutions. If this order relationship remains unchanged, it means that the EN only plays a role in enhancing performance and does not address the core defect of the EMVC method, that is, the FEB problem.

Table 2: Accuracy comparison results with SOTA methods (mean ± standard deviation), where the best performance is highlighted in bold.

| Method | MM-IMDB | | NTU RGB-D | | EgoGesture | |
|---|---|---|---|---|---|---|
| | Modality | F1-W (%) | Modality | Acc (%) | Modality | Acc (%) |
| Uni-view methods | | | | | | |
| Modality 1 | Text (T) | 57.54 | Video (V) | 83.91 | RGB (R) | 93.75 |
| Modality 2 | Image (I) | 49.21 | Pose (P) | 85.24 | Depth (D) | 94.03 |
| Multi-view methods | | | | | | |
| Two-stream (NeurIPS14) | I+T | 60.81 | V+P | 88.60 | - - | - - |
| GMU (ICLR17) | I+T | 61.70 | V+P | 85.80 | - - | - - |
| CentralNet (ECCV18) | I+T | 62.23 | V+P | 89.36 | - - | - - |
| MFAS (CVPR19) | I+T | 62.50 | V+P | 89.50±0.60 | - - | - - |
| MMTM (ICCV20) | - - | - - | V+P | 88.92 | R+D | 93.51 |
| MTUT (3DV19) | - - | - - | - - | - - | R+D | 93.87 |
| 3D-CDC-NAS2 (TIP21) | - - | - - | - - | - - | R+D | 94.38 |
| BM-NAS (AAAI22) | I+T | 62.92±0.03 | V+P | 90.48±0.24 | R+D | 94.96±0.07 |
| DC-NAS (AAAI24) | I+T | 63.70±0.11 | V+P | 90.85±0.05 | R+D | 95.22±0.05 |
| CoMO-NAS (ACMMM24) | I+T | 63.84±0.16 | V+P | 90.94±0.02 | R+D | 95.25±0.03 |
| CSG-NAS (IJCAI24) | I+T | 64.12±0.12 | V+P | 91.12±0.03 | R+D | 95.25±0.04 |
| MCTS-CSG (IJMLC25) | - - | - - | V+P | 91.21±0.10 | R+D | 95.27±0.01 |
| HF-MNAS (TIP25) | I+T | 64.17 | V+P | 91.15 | R+D | **95.31** |
| **EFB-EMVC (ours)** | I+T | **64.53±0.05** | V+P | **91.30±0.03** | R+D | 95.30±0.03 |

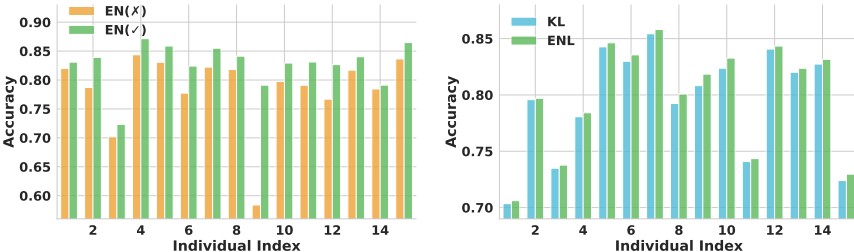

Figure 3: Impact of the EN and the ENL on EFB-EMVC. EN (✗) represents that the final loss function only includes cross-entropy loss $L_{CE}$ and there is no single-view distillation loss $L_{ENL}$.

Experimental results in Table 3 show that EN significantly changes the performance ranking relationships of 15 individuals. By calculating three correlation coefficients of the performance ranking relationships before and after applying EN, results indicate extremely low correlation between the two. Meanwhile, Table 4 systematically presents the distribution of individual pairs with changed performance ranking relationships before and after applying EN among the 15 individuals in the EN ablation experiment. Result shows that within the sample space consisting of 15 individuals, there are 11 pairs of individuals with significantly restructured performance ranking relationships.

The above experimental results fully demonstrate that the distortion effect of FEB on individual performance ranking relationships is extremely prominent, confirming that FEB is a key issue urgently needing to be addressed in the EMVC method. Meanwhile, the role of EN is not limited to performance improvement; instead, it fundamentally corrects the distorted individual performance ranking relationships, enabling EMVC methods to conduct effective optimization based on more accurate performance rankings. This reconstruction of ranking relationships breaks through the constraint bottleneck faced by traditional EMVC methods due to FEB, providing a new solution for algorithm design and theoretical development in this field.

Table 3: Rankings and Correlation Coefficients (PC: Pearson Correlation; SRC: Spearman's Rank Correlation; KT: Kendall's Tau)

| EN | Sort the fitness values of 15 individuals | PC | SRC | KT |
|---|---|---|---|---|
| ✗ | [4, 15, 5, 7, 1, 8, 13, 10, 11, 2, 14, 6, 12, 3, 9] | 0.2143 | 0.2143 | 0.1429 |
| ✓ | [4, 15, 5, 7, 8, 13, 2, 11, 1, 10, 12, 6, 14, 9, 3] | | | |

Table 4: Pairs of individuals with changed fitness value relationships between EN (✗) and EN (✓)

| Individual Pair | EN (✗) | EN (✓) |
|---|---|---|
| [4,0,2,3,-0,-1, -0] vs [0,2,3,4,-2,-3,-0] | 82.00 vs 78.70 | 83.09 vs 83.92 |
| [4,0,2,3,-0,-1, -0] vs [1,0,4,2,-4,-2,-2] | 82.00 vs 81.81 | 83.09 vs 84.12 |
| [4,0,2,3,-0,-1, -0] vs [3,2,4,-3,-4] | 82.00 vs 79.09 | 83.09 vs 83.11 |
| [4,0,2,3,-0,-1, -0] vs [1,4,2,-4,-2] | 82.00 vs 81.71 | 83.09 vs 84.02 |
| [0,2,3,4,-2,-3,-0] vs [0,4,3,-4,-0] | 78.70 vs 79.75 | 83.92 vs 82.93 |
| [0,2,3,4,-2,-3,-0] vs [3,2,4,-3,-4] | 78.70 vs 79.09 | 83.92 vs 83.11 |
| [2,4,-4] vs [0,3,-4] | 70.17 vs 58.39 | 72.33 vs 79.11 |
| [3,4,-3] vs [3,4,0,2,-0,-2,-3] | 77.73 vs 76.69 | 82.41 vs 82.66 |
| [3,4,-3] vs [3,2,-1] | 77.73 vs 78.45 | 82.41 vs 79.12 |
| [0,4,3,-4,-0] vs [3,2,4,-3,-4] | 79.75 vs 79.09 | 82.93 vs 83.11 |
| [3,4,0,2,-0,-2,-3] vs [3,2,-1] | 76.69 vs 78.45 | 82.66 vs 79.12 |

Table 5: Classification accuracy comparison of EDF and DC-NAS before and after applying EN

| Methods | MVoxCeleb | YoutubeFace | NUS | Reuters5 | Reuters3 |
|---|---|---|---|---|---|
| EDF | 93.09±0.20 | 85.83±0.08 | 74.73±0.45 | 81.12±0.25 | 85.49±0.21 |
| **EDF+EN** | **94.70±0.23** | **87.29±0.43** | **75.81±0.71** | **82.72±0.16** | **86.48±0.15** |
| DC-NAS | 92.19±0.07 | 85.28±0.14 | 74.35±0.58 | 81.35±0.28 | 85.86±0.14 |
| **DC-NAS+EN** | **94.82±0.12** | **87.67±0.17** | **75.79±0.76** | **82.66±0.24** | **86.51±0.12** |

**Impact analysis of ENL on EFB-EMVC.** In this part, we will conduct an in-depth analysis of the impacts of the KL divergence and the custom loss function ENL on EFB-EMVC to demonstrate that rich cross-category information is an important component of the KD process. To this end, we have selected the YouTubeFace dataset for the experiment. We first randomly initialize 15 individuals. Subsequently, we perform knowledge KD by applying the KL divergence and the ENL loss function respectively. Conclusions can be drawn from the results on the right side of Fig. 3. After applying the ENL loss function, the accuracy of individuals in the population has been improved to varying degrees. This also fully demonstrates the importance of the ENL loss function in leveraging rich cross-category information.

**Generality of the proposed evolutionary navigator.** In this experiment, we aim to verify whether EN can achieve a high degree of compatibility with other EMVC methods. To this end, we have selected two methods, EDF [7] and DC-NAS [12]. In EDF, individuals are encoded in the form of sequences, while in DC-NAS, individuals are encoded in the form of trees. As shown in Table 5, after integrating EN into these two methods, both methods have achieved significant performance improvements across all datasets. Taking the MVoxCeleb and YoutubeFace datasets as examples, the performance of EDF has been improved by 1.61% and 1.46%, and the performance of DC-NAS has been improved by 2.63% and 2.39%. This experimental result indicates that EN is capable of achieving a high degree of compatibility with various EMVC methods.

## 5 Conclusion

In the field of evolutionary multi-view classification (EMVC), this paper is the first to identify and formally define the fitness evaluation bias (FEB) issue, and systematically analyze its impact on the entire evolutionary process. On this basis, we propose an effective method to alleviate FEB from the perspective of view information content, namely EFB-EMVC. Specifically, EFB-EMVC introduces evolutionary navigators (ENs) into each multi-view model (MVM), and leverages the rich corresponding category and cross-category information contained in ENs to guide the optimization of each view, thereby effectively alleviating the FEB that is prevalent in existing EMVC methods. It is worth emphasizing that ENs can be seamlessly integrated into various multi-view learning methods, demonstrating strong compatibility and extensibility. In future work, there are still several important issues regarding FEB that require in-depth research. For instance: Beyond differences in information content, what other potential factors can cause FEB and what are the corresponding countermeasures? How to integrate efficient optimization techniques to reduce the training cost of EFB-EMVC? These issues will all be key directions for advancing the further development of EMVC.

## Acknowledgements

This work was supported by National Natural Science Foundation of China (Nos. 62306171, T2495251, 62406218, 62506219), the Science and Technology Major Project of Shanxi (No. 202201020101006), and Open Funding Programs of State Key Laboratory of AI Safety (No. 2025-12).

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

# A  Appendix

In the supplemental material:

- **A.1.** Variables and Corresponding Meanings
- **A.2.** Experimental Settings
- **A.3.** Datasets
- **A.4.** Hyperparameter Analysis of EFB-EMVC
- **A.5.** Individuals Used in Ablation Experiment

## A.1  Variables and Corresponding Meanings

To help readers understand the content of this paper more clearly, this section provides a detailed organization and explanation of the variable symbols involved in the paper, with the specific corresponding relationships shown in Table 6.

Table 6: Variable names and their corresponding meanings

| Variables | Meaning |
|---|---|
| $C$ | The number of categories |
| $c$ | The $c$-th category |
| $v$ | The $v$-th view |
| $t$ | The target category |
| $b_c$ | The number of samples assigned to the $c$-th category by the teacher model |
| $\mathcal{V}$ | The number of views contained in the MVM |
| $X_v^c$ | The feature matrix of the $c$-th category in the $v$-th view |
| $K_v^c$ | The kernel matrix of the $c$-th category in the $v$-th view |
| $M_v$ | The mutual information matrix of the $v$-th view |
| $\hat{M}_v$ | The mutual information matrix of the $v$-th view after masking the target category |
| $K$ | The kernel matrix in the Sinkhorn algorithm |
| $\text{HSIC}(K_v^{c_i}, K_v^{c_j})$ | The HSIC score between the $c_i$-th and $c_j$-th category in the $v$-th view |
| $z^{\mathcal{T}}$ | The logits output of the evolutionary navigator |
| $z^{\mathcal{S}}$ | The logits output of the view branch |
| $\hat{z}^{\mathcal{T}}$ | The logits of the evolutionary navigator after masking the target category |
| $\hat{z}^{\mathcal{S}}$ | The logits of the view branch after masking the target category |
| $p^{\mathcal{T}}$ | The probability distribution of the evolutionary navigator |
| $p^{\mathcal{S}}$ | The probability distribution of the view branch |
| $\hat{p}^{\mathcal{T}}$ | The probability distribution of the evolutionary navigator after masking the target category |
| $\hat{p}^{\mathcal{S}}$ | The probability distribution of the view branch after masking the target category |
| $\epsilon$ | The regularization parameter |
| $U, V$ | The iteration matrix |
| $\gamma^{\star}$ | The optimal transport plan |

## A.2  Experimental Settings

In our experiments, all methods are implemented using TensorFlow 2.10.0. The computing environment includes Ubuntu 24.04.2 LTS as the operating system, equipped with an AMD EPYC processor with 160 physical cores (320 logical threads), 566 GB of DDR4 memory, and 8 NVIDIA GeForce RTX 5090 GPUs, each with 32 GB of VRAM. The experimental setup is based on Python 3.9.23 and CUDA 11.2.

**Parameter settings**:

**a) Training of multi-view model (MVM)**: All MVM are trained using the Adam algorithm. The learning rate is set to 0.001, with a first-moment exponential decay rate of 0.9 and a second-moment

exponential decay rate of 0.999. Each network undergoes training for 200 epochs. To prevent overfitting, if the performance of a MVM does not improve after 10 epochs, the training process will be halted.

**b) EFB-EMVC Algorithm**: Inspired by the work of EDF[7], we set the population size to 28, the number of iteration rounds to 20, the crossover rate to 0.9, and the mutation rate to 0.2. For the MVoxCeleb [16] and CB [7] datasets, especially the CB dataset with up to 10,000 categories, directly applying our proposed $\ell_{ENL}$ loss function will lead to an excessively large dimension of the mutual information matrix, resulting in the program being unable to run. In the ablation study section, we have fully verified the key role of cross-category information in the distillation process. To facilitate code reproduction, we use the traditional KL divergence loss instead of $\ell_{ENL}$ on these two datasets to ensure the executability of the method and the stability of the experiment. In addition, for the MM-IMDB [22] dataset — a multi-label and multi-category dataset — the application of the $\ell_{ENL}$ loss function may cause numerical instability during the optimization process due to potential dimension mismatch issues of the mutual information matrix. To address this issue, we still use the traditional KL divergence to replace the $\ell_{ENL}$ loss function. To facilitate the experiment, only the following simplified settings are made for the weights of the loss function: The weight of the fusion output, $\gamma_1$, is set to $1.0\times\mathcal{V}$, where $\mathcal{V}$ represents the number of views of the current individual. For datasets using the $\ell_{ENL}$ loss function, the output weight of each view, $\gamma_2$, is set to 1.0. For the cross-entropy loss between EN and the view branch on the target category, the parameter $\alpha$ is set to 1.0; for the Wasserstein distance between their output distributions on non-target categories, the parameter $\beta$ is set to 0.3. In addition, for datasets using KL divergence, the weight of each view is directly set to 1.0. The temperature coefficient is uniformly set to $t=2$.

### A.3 Datasets

All used datasets in our paper can be downloaded from the `https://github.com/LiShuailzn/Neurips-2025-EFB-EMVC`.

- **MVoxCeleb** [16] is a multi-view audio classification dataset that is constructed with Vox-Celeb dataset [53]. Each audio are extracted five view features and they are two deep feature including ecapa and resnet, and three traditional features including fbank, mfcc and spec. To study aim, Gaussian noise is added on ecapa and resnet mfcc. On this dataset, the specific parameter configuration of our EFB-EMVC method is as follows: the fusion view dimension to 128 and the reuse of view features is not allowed.

- **YouTube-Faces** [19]. The dataset includes 3,425 videos of 1,595 different people downloaded from YouTube. Similar to, we use a subset consists of 101,499 frames of 31 subjects and the same five features are extracted. The parameter configuration of the dataset is as follows: the fusion view dimension to 128, and the reuse of view features is not allowed.

- **NUS-WIDE-128 (NUS)** [20]. NUS dataset contains 43,800 single label images from 128 categories. For each image, six types of image features including color histogram (CH), color correlogram (CORR), edge direction histogram (EDH), wavelet texture (WT), block-wise color moments (CM) and bag of words based on SIFT descriptions (BoW), and one text feature are extracted. The dataset extended from the NUS-WIDE dataset [54]. In our experiments, we use its a subset consisting of 23,438 images from 10 category, including *animal*, *architecture*, *art*, *flowers*, *food*, *man*, *person*, *sky*, *toy*, and *water*. In this subset, each image is related to one label and each category includes at least 1,500 images. For the EFB-EMVC evolutionary algorithm parameters, the fusion view dimension to 128, and the reuse of view features is not allowed.

- **Reuters** [21]. Reuters is a multilingual multi-view dataset, each document is described by five different languages including English, French, German, Spanish and Italian. To make used model can work on this data, the dimensions of all views are reduced to 1,000 using PCA. Then, following, Gaussian noise is added to all views or 3 views for obtaining its two versions, named as Reuters5 [16] and Reuters3 [16], respectively. The parameter configuration of the dataset is as follows: the fusion view dimensions are set to 128, and the reuse of view features is not allowed.

- **CB** [7]. CB dataset designed for chemical structure image recognition in patent retrieval studies, which contains 100,000 chemical structure images distributed into 10,000 categories.

Among them, the specific experimental parameter settings are as follows: the fusion view dimension to 256, and the reuse of view features is not allowed.

- **MM-IMDB** [22]. MM-IMDB dataset for the multi-label film genre classification task, which contains a total of 23 categories. The dataset is divided into a training set of 15,552 films, a validation set of 2,608 films, and a test set of 7,799 films. To ensure fair comparison with other explicitly multi-view fusion approaches, we adopted the same neural network backbone models as BM-NAS and DC-NAS to extract various view features, using weighted F1 score as the evaluation metric. The specific parameter settings are as follows: fusion vector dimension $FD$ is 128, and the reuse of view features is not allowed.

- **NTU** [23]. NTU dataset for multi-view action recognition task containing 60 categories. The training, validation and test sets include 23,760, 2,519 and 16,558 samples, respectively. To ensure the fairness of the experimental results, we followed the data preprocessing pipelines of BM-NAS and DC-NAS. Specifically, we used Inflated ResNet-50 and Co-occurrence as feature extractors for the skeleton and video views. For the EFB-EMVC evolutionary algorithm parameters, the fusion view dimension to 1024, and the reuse of view features is allowed.

- **Ego** [24]. Ego dataset for multi-view gesture recognition task containing 83 categories. The training set of this dataset includes 14,416 samples, the validation set includes 4,768 samples, and the test set includes 4,977 samples. We followed the methods of BM-NAS and DC-NAS, using ResNext-101 as the backbone network for RGB and depth video views. For the EFB-EMVC evolutionary algorithm parameters, the fusion view dimension to 512, and the reuse of view features is not allowed.

### A.4 Hyperparameter Analysis of EFB-EMVC

**Impact Analysis of the fusion dimension on EFB-EMVC.** In this experiment, we selected the YoutubeFace and NUS datasets to investigate the impact of fusion view dimension on the performance of the EFB-EMVC method. Specifically, we first randomly selected 15 individuals as a fixed initial population. Subsequently, within the framework of this fixed initial population, we independently validated the performance of the model when the fusion view dimension was set to 32, 64, 128, 256, and 512 respectively. Experimental results were quantitatively presented using the average performance and maximum performance of the fixed initial population, as shown in Figure 4. It can be observed that on the YoutubeFace dataset, as the fusion dimension increases, the performance of the model shows a trend of continuous improvement and tends to stabilize at high dimensions; whereas on the NUS dataset, increasing the dimension does not bring significant benefits. The above results indicate that the impact of fusion dimension on the performance of the model exhibits dataset dependence: for datasets with higher representation complexity, higher dimensions can effectively enhance feature representation capability, while on datasets with more information redundancy, the benefits of blindly increasing dimensions are limited.

### A.5 Individuals Used in Ablation Experiment

In this section, we will list in detail the individuals used in the ablation experiment part. Specifically, in Table 7, the left side shows the sample set of the EN ablation experiment, and the right side shows the sample set of the ENL ablation experiment.

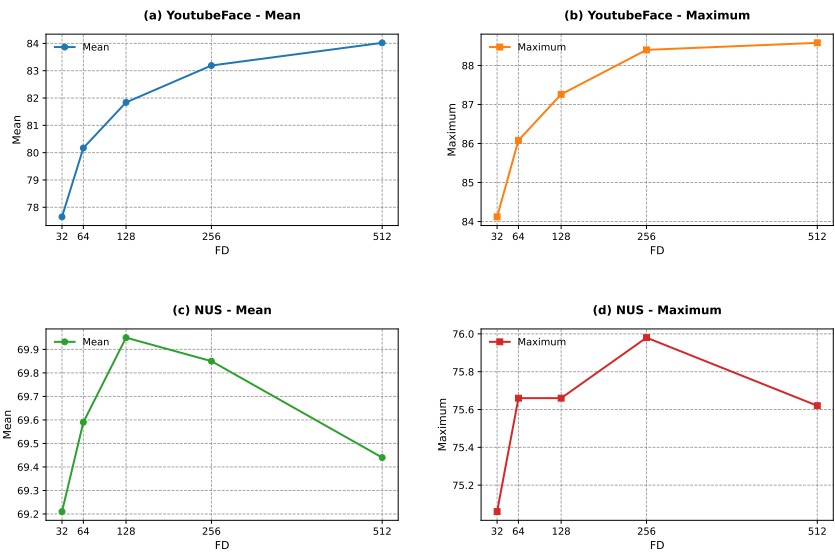

Figure 4: Mean and maximum value variations with integrate the view dimension (FD) on Youtube-Face and NUS datasets

Table 7: The set of individuals used in the ablation experiment

| EN | ENL |
|---|---|
| [4,0,2,3,-0,-1, -0] | [4,0,-4] |
| [0,2,3,4,-2,-3,-0] | [1,0,-2] |
| [2,4,-4] | [4,1,1,-0,-2] |
| [4,3,0,1,2,-1,-1,-2,-4] | [0,0,-3,2,1,-3,-3] |
| [0,2,4,3,1,-1,-2,-3,-2] | [4,4,4,1,-0,-1,-1] |
| [3,4,-3] | [2,4,2,1,-1,-3,-0] |
| [1,4,3,0,2,-1,-4,-2,-3] | [3,4,4,4,1,-2,-4,-3,-0] |
| [1,0,4,2,-4,-2,-2] | [2,2,-4,0,3,0,-1,-0,-3] |
| [0,3,-4] | [0,1,1,-2,-1,2,2,-1,-2] |
| [0,4,3,-4,-0] | [2,0,-0,2,3,3,-0,-0,-2] |
| [3,2,4,-3,-4] | [0,4,4,-4,-1,4,2,-0,-0] |
| [3,4,0,2,-0,-2,-3] | [4,4,2,2,-1,-0,-3] |
| [1,4,2,-4,-2] | [1,4,2,1,-0,-2,-3] |
| [3,2,-1] | [4,1,-3] |
| [3,2,1,4,-4,-1,-2] | [2,4,0,-0,-1] |

