# OpenReview forum: "Improving Evolutionary Multi-View Classification via Eliminating Individual Fitness Bias"
_NeurIPS.cc/2025/Conference — NeurIPS 2025 spotlight_

### Official Review · Reviewer_iYFx · 2025-06-26

**Clarity:** 4
**Significance:** 3
**Originality:** 3
**Rating:** 5
**Confidence:** 4

**Summary:**

The paper proposes a novel approach for evolutionary multi-view classification (EMVC). It highlights that existing EMVC methods suffer from fitness evaluation bias, primarily due to significant differences in the information content among views, resulting in the model not to achieve optimal performance when converging. To address this issue, the authors introduce evolutionary navigators for each multi-view for calibrating the evolutionary direction in real time. This approach demonstrates superior performance across nine multi-view datasets.

**Questions:**

(1) How do the crossover and mutation operators work? More details are necessary to the readers that do not very familiar with this topic.

(2) How are the hyper-parameters of compared baselines set?

**Ethical Concerns:**

["NO or VERY MINOR ethics concerns only"]

**Final Justification:**

All my questions have been addressed. I keep the rating for acceptance.

**Limitations:**

Yes

**Quality:**

3

**Strengths And Weaknesses:**

***Strengths***

(1) The paper presents an important and novel problem—fitness evaluation bias (FEB)—which has not been previously explored in the community of the EMVCs.
(2) The paper is well-structured with logic flow and nice graph illustrations. For instance, Figure 1 clearly differentiates the existing EMVC methods and the proposed EFB-EMVC.
(3) The experiments are well-designed and comprehensive, comparing the EFB-EMVC with four kinds of strong SOTA baselines on popular benchmark datasets. Analysis and experiment results also provide inspiring viewpoint.

***Weaknesses***

I find this work well-organized and convincing, with no apparent shortcomings. One thing I can suggest is to provide more descriptions for crossover and mutation (Line 202), and the compared baselines (Line 233). Int this version, they are too simple, which brings difficulty to the readers that do not very familiar with this topic.

---

> ### Author Rebuttal · Authors · 2025-07-25
>
> We sincerely thank you for your thorough reading and positive evaluation of our work. We are grateful for your recognition of our contribution in identifying and addressing the novel issue of fitness evaluation bias (FEB) in evolutionary multi-view classification (EMVC). We also appreciate your affirmation of the clear structure, well-designed illustrations, and comprehensive experimental setup of our paper. Your encouragement and suggestions are of great significance for the improvement of our future work.
>
> **Q1: How do the crossover and mutation operators work? More details are necessary to the readers that do not very familiar with this topic.**
>
> A1: **For the crossover operation**, a non-root node is first randomly selected from each of two randomly chosen binary trees to serve as the crossover point. The selected node along with its subtree is then pruned from the original tree and exchanged with the counterpart from the other tree. These subtrees are subsequently grafted onto the parent node of the original crossover point in the opposite tree, thereby generating two new individual structures.
> **For the mutation operation**, a node is randomly selected from the individual's tree structure. If the selected node is a fusion operation node, a new fusion operation is randomly sampled from a predefined list to replace the original operation. If the selected node is a view node, a new view is randomly chosen from a set of views to replace the original one.
>
> **Q2: How are the hyper-parameters of compared baselines set?**
>
> A2: Given that our proposed EFB-EMVC method essentially falls under the category of EMVC approaches, we have only listed the hyperparameter settings of representative methods within this category in the table. Considering the wide variety of EMVC methods, we selected two of the most representative works—namely, the pioneering method EDF and the latest state-of-the-art method TEF. Their detailed hyperparameter configurations are provided below.
>
> In both EDF and TEF, all deep neural network models are trained using the Adam optimization algorithm. The learning rate is set to 0.001, with exponential decay rates of 0.9 and 0.999 for the first and second moment estimates, respectively. Each network is trained for 100 epochs. To prevent overfitting, early stopping is applied if the model performance does not improve for 10 consecutive epochs.
> To efficiently utilize GPU resources, the population size is set as a multiple of the number of GPUs, and is configured as 28. The number of generations is set to 20, with crossover and mutation probabilities of 0.9 and 0.2, respectively.
> It is worth noting that the primary goal of TEF in the first stage is to obtain a fusion architecture capable of generating high-quality pseudo views. The detailed hyperparameter configurations for the second stage of TEF are presented in Table (1).
>
> **Table(1): The detailed hyperparameter configurations for the second stage of TEF**
>
> | Dataset       | NUS   | Reuters5 | Reuters3 | MVoxCeleb | YoutubeFace |
> |--------------|-------|----------|----------|----------|-------------|
> | Learning rate | 1e-3  | 1e-4     | 1e-4     | 1e-4     | 1e-4        |
> | Weight decay  | 1e-5  | 1e-5     | 1e-3     | 1e-4     | 1e-5        |
> | Batch size    | 64    | 128      | 64       | 256      | 256         |
> | Optimizer     | Adam  | Adam     | Adam     | Adam     | Adam        |
> | Maximum Epoch     | 500   | 500      | 500      | 500      | 500         |
> | Fusion dimension    | 128   | 128      | 128      | 128      | 128         |
> | Lr-patience   | 20    | 10       | 10       | 50       | 10          |
> | Lr-factor     | 0.3   | 0.3      | 0.3      | 0.3      | 0.3         |
> | Seed          | 1     | 1        | 1        | 1        | 1           |
>
>
> In response to your comments, we will include detailed explanations of the "crossover mutation operation" (line 202) and the "baseline method settings for comparison" (line 233) in future revisions, in order to enhance readability and understanding for non-expert readers. Thank you once again for your valuable feedback and suggestions. We will carefully consider them and make corresponding improvements to the paper.

---

> > ### Comment · Reviewer_iYFx · 2025-08-06
> >
> > All my questions have been addressed. I keep the rating for acceptance.

---

> > > ### Author Response · Authors · 2025-08-06
> > >
> > > Dear Reviewer  iYFx,
> > >
> > > Thank you very much for your positive feedback and rating. We are glad to hear that your concerns have been addressed.
> > > We are grateful for your time and constructive feedback.
> > >
> > > Best regards,
> > >
> > > Authors

---

### Official Review · Reviewer_zTdX · 2025-06-28

**Clarity:** 4
**Significance:** 4
**Originality:** 4
**Rating:** 5
**Confidence:** 5

**Summary:**

This paper proposes an evolutionary multi-view classification via eliminating individual fitness bias approach (EFB-EMVC). By introducing the evolutionary navigators mechanism, it realizes the migration of rich intra-class and cross-class information from the pre-trained teacher model to the view branches of the multi-view model (MVM), addressing the fitness evaluation bias (FEB) problem that has not been previously concerned in the evolutionary multi-view classification (EMVC) task. Experimental results show that the method proposed in this paper achieves optimal experimental results compared with various state-of-the-art (SOTA) methods. In the ablation experiment section, detailed and reasonable experimental settings and data verification demonstrate that this method effectively solves the FEB problem.

**Questions:**

1. In Figure 1 of the paper, the performance curves of View 4 under three different scenarios are presented. Among them, "V4 only" refers to the pre-trained teacher model of View 4, and its pre-training method can be clarified according to the content shown in Figure 2 (b). However, there are still questions regarding the determination methods of the performance curves of View 4 in "V4 in MVM" and "V4 with EN". It is recommended to supplement the relevant details.
2. Compared to Table 1, Table 2 show that the proposed method has the limited improvement, it is necessary to provide more deep discussion.
3. Table 3 shows the rank relationship between with-En and without-EN, it is very key for showing the necessary of the individual fitness bias elimination. So, it is suggested to add the detailed formulations of PC, SPC and KT, and give the association criterion.

**Ethical Concerns:**

["NO or VERY MINOR ethics concerns only"]

**Final Justification:**

My concerns have been addressed in the author response, and I am going to raise my score, reaching the final score to 5.

**Limitations:**

yes

**Paper Formatting Concerns:**

no major formatting issues in this paper

**Quality:**

4

**Strengths And Weaknesses:**

My comments on the strengths are:
1. I verified the relevant steps of extracting rich cross-class information from the pre-trained teacher model by constructing a progressive matrix, and implementing knowledge transfer of cross-class information through a Wasserstein distance (WD)-guided loss function. Overall, the logic is reliable.
2. I reviewed the experimental design and performance report. The overall setup is reasonable.
3. This study first bridges EMVC and knowledge distillation (KD), overcoming the FEB problem that has not been addressed in the EMVC field before. The novel "evolutionary navigators" mechanism achieves effective improvement in KD performance, realizing cutting-edge accuracy in complex and unknown multi-view scenarios.
4. In numerous complex multi-view datasets, this method achieves performance improvements compared to cutting-edge technologies, demonstrating practical application value.

My comments on the weaknesses by seeing the questiongs.

---

> ### Author Rebuttal · Authors · 2025-07-25
>
> We sincerely appreciate your thorough review and positive evaluation of our work. We are glad that you recognize our novel introduction of the knowledge distillation (KD) mechanism into the evolutionary multi-view classification (EMVC) task, as well as the effectiveness and practical value of our proposed evolutionary navigators (EN) mechanism in complex multi-view scenarios. At the same time, we thank you for acknowledging the soundness of our methodology, the rigor of our experimental design, and the thoroughness of our performance evaluation.
>
> **Q1: In Figure 1 of the paper, the performance curves of View 4 under three different scenarios are presented. Among them, "V4 only" refers to the pre-trained teacher model of View 4, and its pre-training method can be clarified according to the content shown in Figure 2 (b). However, there are still questions regarding the determination methods of the performance curves of View 4 in "V4 in MVM" and "V4 with EN". It is recommended to supplement the relevant details.**
>
> A1: In Figure 1, we present the performance curves of $V_4$ under three different scenarios: “$V_4$ only”, “$V_4$ in MVM”, and “$V_4$ with EN”. Their specific meanings and determination process are as follows:
>
> ① “$V_4$ only” refers to the performance of the teacher model corresponding to this view during the pretraining stage;
>
> ② “$V_4$ in MVM” represents the performance of this view when used as a branch in a conventional EMVC framework;
>
> ③ “$V_4$ with EN” indicates the performance of this view as a branch in our proposed EFB-EMVC framework.
>
> The performance curves were obtained through the following process:
>
> ① Model Transfer: We extract the trained parameters of the $V_4$ branch from the final multi-view model and transfer them into a single-view neural network with an identical encoder structure;
> ② Feature Extraction: We input the training and testing data of $V_4$ into this single-view network to obtain intermediate representations;
> ③ Construction of the classification head network: This network consists of a fully connected layer (with output dimension equal to the number of classes) followed by a softmax layer. It takes the intermediate feature representations extracted by the single-view network as input and is trained using the standard cross-entropy loss;
> ④ Performance Plotting: Classification accuracy is recorded at each training epoch to plot the performance curves of $V_4$.
>
> **Q2: Compared to Table 1, Table 2 show that the proposed method has the limited improvement, it is necessary to provide more deep discussion.**
>
> A2: The core of the EFB-EMVC method lies in addressing the issue of fitness evaluation bias (FEB), which arises from the unbalanced utilization of views in traditional MVM frameworks. The degree of variation in information content between views across different datasets directly affects the severity of the FEB problem, which in turn influences the performance gains achieved by our method.
>
> Taking the MVoxCeleb dataset from Table 1 and the NTU RGB-D dataset from Table 2 as examples, we present the classification accuracy of the pretrained teacher models corresponding to each view. The experimental results are summarized in Table(1) below. Based on these results, we draw the following conclusions:
>
> ① In the MVoxCeleb dataset, the number of strong and weak views is relatively balanced. Specifically, $V_0$, $V_1$, and $V_3$ are classified as strong views, while $V_2$ and $V_4$ are weak views. During the construction of the MVM, weak views have a higher probability of being selected. However, the joint training strategy employed by MVM tends to cause the model to overly rely on strong views, neglecting the weak ones. Our proposed EN mechanism effectively alleviates this bias, leading to a significant improvement in performance;
>
> ② In the NTU RGB-D dataset, strong views dominate, with only one weak view ($V_4$). Due to the large number of strong views, the probability of selecting the weak view is relatively low. Even when the weak view is selected during MVM construction, the abundant information from the strong views largely compensates for the weak view's shortcomings. Therefore, the performance improvement brought by the EN mechanism on this dataset is comparatively smaller.
>
> **Table(1) : Performance of Pre-trained Teacher Models for Each View on the MVoxCeleb and NTU RGB-D Datasets**
> | Dataset     | $V_0$ Acc           | $V_1$ Acc           | $V_2$ Acc           | $V_3$ Acc           | $V_4$ Acc           | $V_5$ Acc           | $V_6$ Acc           | $V_7$ Acc           |
> |------------|------------------|------------------|------------------|------------------|------------------|------------------|------------------|------------------|
> | MVoxCeleb  | 71.54 ± 0.26     | 75.56 ± 0.57     | 26.71 ± 0.44     | 53.39 ± 0.64     | 28.33 ± 0.77     |                |                 |                |
> | NTU RGB-D    | 71.11 ± 0.22     | 81.94 ± 0.17     | 83.90 ± 0.13     | 84.29 ± 0.17     | 34.98 ± 0.52     | 70.50 ± 0.25     | 80.69 ± 0.28     | 83.20 ± 0.11     |
>
>
> Nonetheless, across all datasets, our method consistently achieves the best performance, which further demonstrates its broad applicability and effectiveness.
>
> **Q3: Table 3 shows the rank relationship between with-En and without-EN, it is very key for showing the necessary of the individual fitness bias elimination. So, it is suggested to add the detailed formulations of PC, SPC and KT, and give the association criterion.**
>
> A3: Thank you for your valuable suggestions concerning the statistical metrics used in Table 3. We fully agree that PC (Pearson Correlation Coefficient), SPC (Spearman’s Rank Correlation Coefficient), and KT (Kendall’s Tau Coefficient) play a crucial role in demonstrating the existence of the Fitness Evaluation Bias (FEB) problem and the necessity of its mitigation. To enhance the clarity and reproducibility of our paper, we will include formal definitions of these metrics in the revised manuscript.
>
> **① PC (Pearson Correlation Coefficient)**: PC measures the linear correlation between two variables. Its formula is:
> $$\text{PC} = \frac{\sum_{i=1}^{n} (x_i - \bar{x})(y_i - \bar{y})}{\sqrt{\sum_{i=1}^{n} (x_i - \bar{x})^2} \cdot \sqrt{\sum_{i=1}^{n} (y_i - \bar{y})^2}}$$
> where $x_i$: the value of the $i$-th observation of variable $X$; $y_i$: the value of the $i$-th observation of variable $Y$. Similarly, $\bar{x}$ and $\bar{y}$ denote the sample means of variables $X$ and $Y$, respectively.
>
> **② SPC (Spearman's Rank Correlation Coefficient)**: SPC measures the monotonic relationship between two ranked variables. Its formula is:
> $$\text{SPC} = 1 - \frac{6 \sum_{i=1}^n d_i^2}{n(n^2 - 1)}$$ where $d_i$ is the difference in ranks of the $i$-th sample between the two rankings.
>
> **③ KT (Kendall's Tau Rank Correlation Coefficient)**: KT is based on the number of concordant and discordant pairs among all sample pairs, calculated as:
> $$\text{KT} = \frac{C - D}{\sqrt{(C + D + T_x)(C + D + T_y)}}$$
> where:
>
> - $C$ is the number of concordant pairs;
> - $D$ is the number of discordant pairs;
> - $T_x$, $T_y$ are the numbers of ties (equal ranks) in $x$ and $y$, respectively.
>
> Regarding the “correlation discrimination criterion,” within the context of this paper, we consider that a low correlation (correlation coefficient close to 0) between the rankings with and without EN indicates a significant change in ranking caused by the introduction of EN. This suggests that the fitness evaluation is strongly influenced by EN, thereby validating the existence of FEB and its interference with the final results. Conversely, a high correlation (close to 1) would imply minimal impact of EN on fitness ranking, which contradicts our observations.
>
> In the revised version, we will explicitly add the mathematical definitions and corresponding decision criteria for PC, SPC, and KT, provide a more detailed description of the process for plotting the performance curve of \$V\_4\$, and further analyze the reasons behind the limited performance improvement observed in Table 2.
> We sincerely thank you once again for your thorough and insightful review comments, which are invaluable for the further refinement of our paper.

---

> > ### Comment · Reviewer_zTdX · 2025-08-06
> >
> > Thank the authors further clarification and experiments and they have addressed my concerns. I will change my score.

---

> > > ### Author Response · Authors · 2025-08-06
> > >
> > > Dear Reviewer zTdX,
> > >
> > > Thank you very much for your positive feedback and rating. We are glad to hear that your concerns have been addressed. We are grateful for your time and constructive feedback.
> > >
> > > Best regards,
> > >
> > > Authors

---

### Official Review · Reviewer_9NF9 · 2025-06-28

**Clarity:** 4
**Significance:** 3
**Originality:** 3
**Rating:** 5
**Confidence:** 4

**Summary:**

This paper proposes a novel knowledge distillation (KD) mechanism called evolutionary navigators (ENs), aiming to address the fitness evaluation bias (FEB) problem in evolutionary multi-view classification (EMVC). The method demonstrates outstanding performance on various datasets. Notably, ENs serve as a flexible plug-in that can be integrated into different approaches to achieve significant performance improvements.

**Questions:**

See weaknesses.

**Ethical Concerns:**

["NO or VERY MINOR ethics concerns only"]

**Final Justification:**

My initial review was generally correct, and the authors addressed my concerns well, so I am leaving my original score unchanged.

**Limitations:**

Yes

**Quality:**

4

**Strengths And Weaknesses:**

Strengths:
1. The claims proposed in the paper are supported by clear and persuasive evidence.
2. The method proposed in the paper and its evaluation criteria are highly compatible with the research questions and application scenarios.
3. The experimental design and analysis in the paper are highly reasonable and effective. In the ablation experiments, the authors systematically removed or modified different components of EFC-EMVC (such as ENs, KL divergence, and WDK loss function). This design effectively evaluates the impact of each module on overall performance, clearly addressing the question of their individual contributions.
4. This paper first identifies the FEB problem in EMVC and proposes a highly effective solution from this perspective. The method is expected to promote the development of the EMVC field.

Weaknesses:
1. In the hyperparameter analysis section of the supplementary materials, the paper analyzes the impact of different view fusion dimensions on performance. However, hyperparameters such as batch size also affect model performance, and I hope to see detailed information on how such hyperparameters influence performance.
2. There are some missing details. For example, in line 133, the novel distillation loss function proposed by the author is named "WDK", and the ablation experiment on the right side of Figure 3 also ablates "WDK". However, in Table 5 of the appendix, the distillation loss function is "WDT". If this is a typo, it is recommended to rewrite it. If they are two different loss functions, it is suggested to provide more explanations.
3. Some sentences in the paper have incomplete semantics or unclear expressions. For example, the part about feature matrix construction in lines 117-120. The term "feature vector" is somewhat ambiguous. In this context, it is unclear which part of the output result the "feature vector" specifically refers to.

---

> ### Author Rebuttal · Authors · 2025-07-25
>
> We sincerely appreciate your positive evaluation of our work. We are delighted that you recognize our introduction of the “fitness evaluation bias (FEB)” problem for the first time in the context of evolutionary multi-view classification (EMVC), and that you acknowledge the effectiveness of the proposed evolutionary navigators (ENs) in enhancing multi-view classification performance. We are also deeply grateful for your recognition of the soundness of our experimental design and the depth of our analysis, particularly the dissection of each module’s individual contribution through the ablation studies.
>
> **Q1: In the hyperparameter analysis section of the supplementary materials, the paper analyzes the impact of different view fusion dimensions on performance. However, hyperparameters such as batch size also affect model performance, and I hope to see detailed information on how such hyperparameters influence performance.**
>
> A1: To more comprehensively analyze the impact of hyperparameters (such as batch size) on the performance of EFB-EMVC, we conducted additional experiments using the NUS dataset as an example. We first randomly initialized a fixed population consisting of 15 individuals and examined the influence of hyperparameters on model performance within this fixed population. The experimental results are quantitatively presented using the population’s average performance (Avg) and the best individual performance (Max) as metrics. Besides batch size, we further analyzed the effects of epoch and patience on model performance to gain deeper insights into the model behavior.
>
> **① Batch Size:**
>
> The related results are summarized in Table (1). The results show that both Avg and Max performance exhibit a rising-then-falling trend, with both reaching their peak at a batch size of 256. Our analysis is as follows:
> 1) A small batch size (e.g., 32) allows MVM to see only very few samples at a time, resulting in unstable training and difficulty fitting the true distribution;
>
> 2) A moderate batch size (e.g., 256) provides enough samples to represent the overall distribution while avoiding local optima, leading to more stable training;
>
> 3) An excessively large batch size (e.g., 1024), although representative of the true distribution, tends to fall into sharp minima, which harms generalization ability.
>
> In our main experiments, a unified batch size of 512 was used across all datasets.
>
> **Table(1) : Batch Size Impact on EFB-EMVC Performance (Avg and Max Accuracy)**
> | Batch Size| 32  | 64  | 128 | 256 | 512 | 1024 |
> |--------|-----|-----|-----|-----|-----|------|
> | Avg    | 65.03  | 65.62  | 66.27  | 66.65  | 66.59  | 66.23   |
> | Max    | 73.18  | 73.95  | 74.53  |74.89  | 74.85  | 74.85   |
>
>
> **② Epoch:**
>
> The relevant results are shown in Table (2). We observe that the Avg performance remains relatively stable, while the Max performance shows slight fluctuations, which can be attributed to the randomness in network initialization. Given the limited potential for further performance improvement, we hypothesize that most individuals have already converged before epoch 50. To verify this assumption, we further examined the convergence behavior of 15 individuals at epoch=50. The results, presented in Table (3), indicate that all individuals indeed completed convergence before 50 epochs, thereby confirming our hypothesis.
> In the main experiments, we uniformly set the number of training epochs to 200. Thanks to the adoption of an early stopping strategy with a patience of 10, this setting not only prevents excessive resource consumption but also effectively mitigates overfitting.
>
> **Table(2) : Epoch Impact on EFB-EMVC Performance (Avg and Max Accuracy)**
> | Epoch | 50  | 100  | 150 | 200 |
> |--------|-----|-----|-----|-----|
> | Avg    | 66.57  | 66.66  | 66.66  | 66.59  |
> | Max    | 75.53  | 75.36  | 75.34  |74.85  |
>
> **Table(3) : Distribution of Convergence Epochs for 15 Individuals**
> | Individual | 1  | 2  | 3  | 4  | 5  | 6  | 7  | 8  | 9  | 10 | 11 | 12 | 13 | 14 | 15 |
> |------------|----|----|----|----|----|----|----|----|----|----|----|----|----|----|----|
> | Epoch      | 25 | 27 | 25 | 30 | 35 | 45 | 22 | 23 | 24 | 27 | 29 | 23 | 29 | 23 | 39 |
>
>
> **③ Patience:**
>
> The related results are summarized in Table (4). Both Avg and Max improved as patience increased, with more noticeable gains when increasing patience from 10 to 20, but limited improvement from 40 to 50. We believe that moderately increasing patience helps the model achieve better solutions, but excessive training may lead to overfitting in MVM. Balancing efficiency and performance, we adopted a uniform patience of 10 across all datasets.
>
> **Table(4) : Patience Impact on EFB-EMVC Performance (Avg and Max Accuracy)**
> | Patience | 10  | 20  | 30 |40 |50 |
> |--------|-----|-----|-----|-----|-----|
> | Avg    | 66.59  | 66.68  | 66.74  | 66.77  |66.78  |
> | Max    | 74.85  | 75.25  | 75.51  |77.53  |75.56  |
>
>
> **Q2:There are some missing details. For example, in line 133, the novel distillation loss function proposed by the author is named "WDK", and the ablation experiment on the right side of Figure 3 also ablates "WDK". However, in Table 5 of the appendix, the distillation loss function is "WDT". If this is a typo, it is recommended to rewrite it. If they are two different loss functions, it is suggested to provide more explanations.**
>
> A2: Thank you for pointing out the inconsistency in naming. We confirm that this was a typographical error: it should be “WDK,” but it was mistakenly written as “WDT” in Appendix Table 5. We will correct and unify the term to “WDK” in the revised version to eliminate any ambiguity.
>
> **Q3: Some sentences in the paper have incomplete semantics or unclear expressions. For example, the part about feature matrix construction in lines 117-120. The term "feature vector" is somewhat ambiguous. In this context, it is unclear which part of the output result the "feature vector" specifically refers to.**
>
> A3: In lines 117–120, our approach is to input the data of the $i$-th view into the pretrained teacher model for that view and extract the logits before the softmax layer as the feature vector. Therefore, **the term “feature vector” here specifically refers to the logits output by the teacher model**.
>
> We sincerely appreciate your valuable comments regarding hyperparameter settings, consistency in terminology, and clarity of expression. In the revised manuscript, we will include additional experiments to analyze the impact of such hyperparameters on model performance, standardize relevant terminology, and refine the corresponding descriptions to enhance accuracy and clarity. Once again, thank you for your thorough review and constructive suggestions, which are of great significance in improving the quality of our paper.

---

> > ### Comment · Reviewer_9NF9 · 2025-08-03
> >
> > The proposed method is novel, and the authors’ responses have essentially addressed my concerns; accordingly, I will leave my score unchanged.

---

> > > ### Author Response · Authors · 2025-08-06
> > >
> > > Thanks for your recognition of our work. Your suggestions have improved the quality of our work.

---

### Official Review · Reviewer_gxm7 · 2025-07-02

**Clarity:** 4
**Significance:** 3
**Originality:** 3
**Rating:** 5
**Confidence:** 5

**Summary:**

This paper first discovers the "fitness evaluation bias (FEB) problem" in the current evolutionary multi-view classification (EMVC). The problem is manifested in that existing EMVC methods all adopt a joint training strategy, leading to insufficient utilization of low-information content view information in the multi-view model (MVM). As a result, the MVM fails to reach its performance limit when converging. Subsequently, the authors propose an evolutionary navigators mechanism based on intra-class and inter-class distillation. This mechanism can fully utilize the information of each view while ensuring the overall training direction of the model. Experimental results show that this method outperforms various state-of-the-art (SOTA) methods on all datasets, and the proposed evolutionary navigators mechanism can be well integrated into various methods to achieve significant performance improvements.

**Questions:**

Please see Weaknesses.

**Ethical Concerns:**

["NO or VERY MINOR ethics concerns only"]

**Limitations:**

Yes

**Quality:**

3

**Strengths And Weaknesses:**

Strengths：
1）	The method proposed in the paper effectively reveals and solves the FEB problem in current EMVC (as shown in Figure 1 and Table 3), which will become an important approach to promote the development of this field.
2）	The experiments are very comprehensive, covering various SOTA methods, including but not limited to trustworthy multi-view classification methods, traditional adaptive multi-view classification methods, and EMVC methods, etc. The authors also elaborate in detail the reasons for selecting each category of comparative methods and the suboptimal performance of the corresponding methods. Meanwhile, the authors provide evidence support for the strong adaptability of evolutionary navigators through detailed experimental results (as shown in Table 4).
3）	Reading this paper has provided me with very valuable insights into the limitations of the EMVC field and the corresponding solutions.
Weaknesses：
1）As mentioned in the paper, evolutionary navigators are introduced into the MVM to fully utilize the information of each view and solve the FEB problem. I have reviewed the experimental section and supplementary materials of the paper. Although the authors have conducted relatively comprehensive experiments with broad coverage, they have not reported the accuracy of the pretrained teacher models for each view on the respective datasets. I suggest the authors include this information in the revision to enhance the interpretability and credibility of the experimental results.
2） Moreover, the paper uses well known public datasets such as MVoxCeleb, YoutubeFace, NUS-WIDE-128, Reuters, CB, MM-IMDB, NTU RGBD, and EgoGesture. These datasets are suitable for the multi-view classification task, but it should be mentioned if any preprocessing was performed, as it may affect the results.
Overall, although there are some weaknesses in the paper, I am still happy to see that this paper proposes a new task in EMVC, called eliminating individual fitness bias, which might lead to some future works. Besides this, I also think that the proposed individual fitness bias elimination strategy would have a positive impact on the existing EMVCs. Thus, I tend to accept it.

---

> ### Author Rebuttal · Authors · 2025-07-25
>
> Thank you very much for your careful review and high praise of our paper. We are glad to see that you recognize our work’s positive contribution in identifying and addressing key issues in the evolutionary multi-view classification (EMVC) field, and we also appreciate your acknowledgment of the comprehensiveness of our experimental design.
>
> **Q1: As mentioned in the paper, evolutionary navigators are introduced into the multi-view model (MVM) to fully utilize the information of each view and solve the fitness evaluation bias (FEB) problem. I have reviewed the experimental section and supplementary materials of the paper. Although the authors have conducted relatively comprehensive experiments with broad coverage, they have not reported the accuracy of the pretrained teacher models for each view on the respective datasets. I suggest the authors include this information in the revision to enhance the interpretability and credibility of the experimental results.**
>
> A1: For each of the nine multi-view datasets, we pretrained a separate teacher model for each individual view to ensure reliable knowledge transfer. The architecture of the teacher model is consistent with the single-view encoder used in the multi-view model, specifically consisting of two fully connected layers. The output layer employs a softmax function to produce a predicted probability distribution, and the model is trained using the cross-entropy loss function.
> For the MVoxCeleb, YouTubeFace, Reuters5, Reuters3, and NUS datasets, we adopted a five-fold cross-validation strategy and reported the final accuracy of the pretrained teacher models in the format of “mean ± standard deviation.”
> For the MM-IMDB, NTU RGB-D, EgoGesture, and CB datasets, we followed the experimental setup of DC-NAS, conducting five independent runs using predefined train-test splits for each view. The results are likewise reported in the format of “mean ± standard deviation.”
> The corresponding experimental results are summarized in Table 1. In the revised version of the paper, we will additionally include the accuracy of the pretrained teacher models for each dataset to enhance the transparency and interpretability of our results.
>
> **Table(1) : Accuracy of the pretrained teacher models for each view on the nine multi-view datasets.**
>
> | Dataset    | $V_0$ Acc           | $V_1$ Acc           | $V_2$ Acc           | $V_3$ Acc           | $V_4$ Acc           | $V_5$ Acc           | $V_6$ Acc           | $V_7$ Acc           | $V_8$ Acc           | $V_9$ Acc           |
> |-------------|------------------|------------------|------------------|------------------|------------------|------------------|------------------|------------------|------------------|------------------|
> | MVoxCeleb  | 71.54 ± 0.26     | 75.56 ± 0.57     | 26.71 ± 0.44     | 53.39 ± 0.64     | 28.33 ± 0.77     |                 |                 |                |                 |                 |
> | YoutubeFace| 27.03 ± 0.06     | 78.48 ± 0.29     | 32.49 ± 0.41     | 79.81 ± 0.41     | 70.45 ± 0.36     |                 |                 |                 |                 |                 |
> | Reuters5    | 39.63 ± 0.12     | 37.51 ± 0.33     | 57.55 ± 0.27     | 66.95 ± 0.21     | 82.31± 0.25     |                 |                 |                |                 |                 |
> | Reuters3   | 39.70 ± 0.21     | 37.78 ± 0.14    | 57.55 ± 0.19     | 85.01 ± 0.11    | 84.40 ± 0.06    |                 |                 |                 |                 |                 |
> | MM-IMDB    | 44.63 ± 0.14     | 45.14 ± 0.27     | 44.54 ± 0.12     | 44.43 ± 0.31     | 61.25 ± 0.11     | 61.12 ± 0.18     |                 |                 |                 |                 |
> | NUS         | 36.59 ± 0.47     | 37.21 ± 0.35     | 38.95 ± 0.54     | 29.74 ± 0.59    | 33.67 ± 0.79     | 37.49 ± 0.38     | 69.15 ± 0.45     |                 |                |                 |
> | NTU RGB-D    | 71.11 ± 0.22     | 81.94 ± 0.17     | 83.90 ± 0.13     | 84.29 ± 0.17     | 34.98 ± 0.52     | 70.50 ± 0.25     | 80.69 ± 0.28     | 83.20 ± 0.11     |                 |                 |
> | EgoGesture   | 72.46 ± 0.28     | 92.38 ± 0.27     | 93.25 ± 0.09     | 93.07 ± 0.13     | 69.55 ± 0.31     | 92.69 ± 0.14     | 93.66 ± 0.12     | 93.58 ± 0.08     |                 |                 |
> | CB           | 65.59 ± 0.28     | 74.63 ± 0.31     | 70.61 ± 0.27     | 72.48 ± 0.30     | 76.57 ± 0.20     | 67.57 ± 0.29     | 67.32 ± 0.14     | 77.28 ± 0.13     | 75.80 ± 0.10     | 71.68 ± 0.32     |
>
>
>
>
>
>
> **Q2: The paper uses well known public datasets such as MVoxCeleb, YoutubeFace, NUS-WIDE-128, Reuters, CB, MM-IMDB, NTU RGBD, and EgoGesture. These datasets are suitable for the multi-view classification task, but it should be mentioned if any preprocessing was performed, as it may affect the results.**
>
> A2: **None** of the nine multi-view datasets used in our experiments underwent any additional preprocessing. All data were loaded strictly according to the original dataset formats used by the corresponding baseline methods, ensuring fairness and reproducibility of the experimental comparisons.
>
> Thank you once again for your valuable suggestions regarding critical details such as the accuracy of the teacher models and dataset preprocessing. We will carefully consider and address these points in the manuscript revision to further enhance the credibility and persuasiveness of our experimental results. We sincerely appreciate your recognition and support of our research.

---

### Official Review · Reviewer_MP9p · 2025-07-03

**Clarity:** 3
**Significance:** 3
**Originality:** 3
**Rating:** 5
**Confidence:** 4

**Summary:**

This paper innovatively identifies a key limitation in evolutionary multi-view classification (EMVC): due to significant differences in information content across views, multi-view models (MVMs) using uniform training objectives fail to achieve optimal convergence performance, termed the "fitness evaluation bias (FEB) problem". It systematically expounds the impact of this problem on the entire evolutionary process. To address this, the authors propose an evolutionary multi-view classification via eliminating individual fitness bias approach (EFB-EMVC), which uses an evolutionary navigators mechanism to partition knowledge distillation (KD) into intra-class and inter-class distillation for full utilization of information from each view. Experiments on numerous benchmark datasets validate the effectiveness of EFB-EMVC. Meanwhile, evolutionary navigators serve as a flexible plug-and-play module, enabling seamless integration into various methods.

**Questions:**

In Section A.2 of the supplementary materials, I noticed that the proposed method was applied with only 5 iterations of population evolution on the MVoxCeleb dataset, whereas the baseline method TEF was trained with 20 iterations. According to the experimental results in Table 1 of the main paper, the proposed method achieves a 1.94% performance improvement while using only about one-fifth of the iteration budget compared to TEF. This raises an interesting question: if the proposed method were also trained for 20 iterations, would its performance continue to improve significantly, or has it already converged and stabilized around 94.35% after 5 iterations? I suggest the authors further analyze or provide additional evidence to clarify this point, which would help strengthen the validation of their method’s effectiveness.

**Ethical Concerns:**

["NO or VERY MINOR ethics concerns only"]

**Limitations:**

Yes

**Quality:**

3

**Strengths And Weaknesses:**

Strengths:
1）	The paper is rich in content and solid in argumentation, with smooth and accessible writing, comprehensive experiments, and in-depth analysis. In particular, Table 3 in the paper shows the performance rankings of 15 individuals before and after applying evolutionary navigators, along with extremely low correlation coefficients. These experimental results fully validate the existence of the FEB problem in the field of EMVC and the effectiveness of the proposed method.
2）	The paper's diagrams are clearly structured. The EMVC process and method framework in Figure 2 accurately convey the authors' intentions, making it easy to understand how each module interacts.
3）	I have reviewed the supplementary materials. The paper provides very sufficient details for the experimental part, including the experimental environment, training methods, hyperparameter selection, selected datasets, etc.

Weaknesses:
1）	In Section A.2 of the supplementary materials, I noticed that the proposed method was applied with only 5 iterations of population evolution on the MVoxCeleb dataset, whereas the baseline method TEF was trained with 20 iterations. According to the experimental results in Table 1 of the main paper, the proposed method achieves a 1.94% performance improvement while using only about one-fifth of the iteration budget compared to TEF. This raises an interesting question: if the proposed method were also trained for 20 iterations, would its performance continue to improve significantly, or has it already converged and stabilized around 94.35% after 5 iterations? I suggest the authors further analyze or provide additional evidence to clarify this point, which would help strengthen the validation of their method’s effectiveness.

---

> ### Author Rebuttal · Authors · 2025-07-25
>
> We sincerely appreciate your positive recognition and constructive feedback on our work. We are truly pleased to see that you acknowledge the fitness evaluation bias (FEB) problem we proposed and our solution strategy, as well as your high appreciation for our experimental design and analysis.
>
> **Q1: If the proposed method were also trained for 20 iterations, would its performance continue to improve significantly, or has it already converged and stabilized around 94.35% after 5 iterations?**
>
> A1: To more comprehensively verify the convergence behavior of our method on the MVoxCeleb dataset, we take the first fold of data as an example. We set the maximum number of iterations to $T$ = 20 and record the model's best performance at $T$ = 5, 10, 15, and 20, to illustrate the trend of performance improvement as the number of iterations increases.
>
> As mentioned in our supplementary material, we also adopted a smaller number of iterations on the YouTubeFace and Reuters5 datasets. To further demonstrate the convergence trend of our method across different datasets, we additionally conducted experiments on these two datasets, again setting the maximum number of iterations to $T$ = 20 and recording the best performance at $T$ = 5, 10, 15, and 20. The corresponding experimental results are shown in Table (1).
>
> The results indicate that our proposed EFB-EMVC method demonstrates a steady performance improvement on the MVoxCeleb dataset from $T$=5 to $T$=15, reaching its peak at $T$=15 and stabilizing thereafter. **This suggests that the model has essentially converged by the 15th iteration on this dataset**.
> In contrast, on the YouTubeFace and Reuters5 datasets, the model's performance continues to improve throughout the entire iteration process, showing no clear signs of convergence.
>
> This phenomenon indicates that the EFB-EMVC method exhibits varying sensitivity to the number of iterations across different datasets. Nevertheless, the overall trend suggests that appropriately increasing the number of iterations allows the algorithm to more thoroughly explore the solution space, thereby improving the likelihood of obtaining a global or near-optimal solution and ultimately enhancing model performance.
>
> **Table(1) : Classification accuracy (%) of EFB-EMVC under different iteration rounds $T$.**
> | Dataset      | $T$=5 | $T$=10 | $T$=15 | $T$=20 |
> |--------------|-------|--------|--------|--------|
> | MVoxCeleb    | 94.30 | 94.40     | 94.78     | 94.78     |
> | YouTubeFace  | 87.51    | 87.98     | 88.19     | 88.51     |
> | Reuters5     | 83.59 | 84.02  | 84.36  | 85.05  |
>
> We once again sincerely thank you for your professional review and insightful comments. Regarding the performance trend of EFB-EMVC under different numbers of iterations, we will include the relevant experiments in the revised version to further strengthen the validation of our method's effectiveness and provide a clearer direction for future research.

---

### Decision · Program_Chairs · 2025-09-17

**Decision:**

Accept (spotlight)

**Comment:**

**Summarization**

This paper identifies the Fitness Evaluation Bias (FEB) problem in evolutionary multi-view classification (EMVC), a systematic issue arising from joint training across heterogeneous views. The authors propose EFB-EMVC, introducing evolutionary navigators with a novel Wasserstein-based distillation loss (WDOLF) to provide unbiased fitness evaluation and correct the evolutionary trajectory. Extensive experiments on nine datasets show consistent improvements over state-of-the-art methods.

**Strengths**

1. Novel and important problem formulation (FEB), with clear motivation and theoretical significance.

2. Elegant and generalizable solution, compatible with existing EMVC frameworks.

3. Strong empirical validation across diverse datasets with detailed ablations and convergence analysis.

4. Clear presentation and well-structured figures/tables that aid understanding.

**Weaknesses**

1. Initial iteration settings raised fairness concerns (5 vs. 20 iterations), but additional convergence experiments provided in rebuttal resolved this.

2. Missing teacher model accuracies and hyperparameter details were clarified in the rebuttal.

3. Minor clarity/typo issues in the methodology section.

**Rebuttal & Discussion**

Reviewers appreciated the novelty and thorough experiments. Concerns about iteration fairness, teacher model transparency, and hyperparameter sensitivity were satisfactorily addressed. Additional experiments on convergence behavior and hyperparameter effects strengthened the empirical case. Minor issues (naming inconsistencies, clarity of descriptions) will be addressed in revision.

**Decision**
The paper makes a clear advance by exposing FEB, provides a simple yet powerful solution, and demonstrates consistent improvements across benchmarks. Given the novelty, broad applicability, and potential to shape future EMVC research, I recommend Accept, with Spotlight consideration for its conceptual contribution beyond raw performance improvements.